# The Long-Term Efficacy and Sustainability of the Tabby Improved Prevention and Intervention Program in Reducing Cyberbullying and Cybervictimization

**DOI:** 10.3390/ijerph20085436

**Published:** 2023-04-07

**Authors:** Anna Sorrentino, Francesco Sulla, Margherita Santamato, Annarosa Cipriano, Stefania Cella

**Affiliations:** 1Department of Psychology, University of Campania “Luigi Vanvitelli”, 81100 Caserta, Italy; 2Department of Human Studies, University of Foggia, 71121 Foggia, Italy; 3Observatory on Eating Disorders, Department of Psychology, University of Campania “Luigi Vanvitelli”, 81100 Caserta, Italy

**Keywords:** long-term assessment, prevention programs sustainability, cyberbullying, cybervictimization

## Abstract

Although cyberbullying and cybervictimization prevention programs have proved effective in the short term, their effectiveness remains unclear in the long run. Thus, the present study evaluated the long-term effects of the Tabby Improved Prevention and Intervention Program (TIPIP). Participants were 475 middle and high school students (Mage = 12.38; SD = 1.45; F = 241, 51%), of whom, 167 were in the Experimental Group (EG; Mage = 13.15; SD = 1.52; M = 51.5%), and 308 were in the Control Group (CG; Mage = 13.47; SD = 1.35; M = 47.7%). Students completed measures assessing cyberbullying and cybervictimization at three time points: baseline (T1), immediately after the intervention (6 months, T2), and at 1 year (T3). The results showed no significant effects of the TIPIP in reducing both cyberbullying and cybervictimization over time. Overall, our results confirm the lack of effectiveness of long-term preventive programs and emphasize that different curricula should be implemented in future programs to prevent and manage cyberbullying and cybervictimization, also taking into account psychological mechanisms and processes involved in such behaviors.

## 1. Introduction

The increasing spread of digital devices and the Internet, especially among youth, has developed new social relationships and behaviors [1]. However, for its nature, such a highly connected world may have negative consequences, and cyberspace has been implicated as a new risky environment for bullying [2,3]. Over the past decades, the increasing use of Information and Communications Technologies (ICT) to intentionally and repeatedly harass, humiliate, and denigrate victims who cannot easily defend themselves [4] have emerged as a major public health issue affecting adolescents’ well-being worldwide [5].

Over the past decades, research has examined the prevalence and consequences of cyberbullying to unravel the complexity of the phenomenon.

Given the heterogeneity and the relative lack of consistency in the definition and methodological differences in existing epidemiological studies (e.g., measurement tools, target population, time span period, research methodology), prevalence rates are highly inconsistent [2,6,7,8].

For example, examining 159 prevalence studies, Brochado et al. [9] have found that the prevalence of cybervictimization ranged from 1.0 to 61.0%, while cyberbullying ranged from 3.0 to 39.0% in the previous year. Accordingly, a recent systematic review examining the worldwide prevalence of cyberbullying and cybervictimization [5] reported an average rate of 25.03% (range = 6.0–46.3%) for cyberbullying and 33.08% (range = 13.99–57.3%) for cybervictimization. Similar variability was found in the lifetime prevalence rate, ranging from 1.2% to 44.1% for perpetration and 4.9% to 65.0% for victimization [9].

Despite the high heterogeneity in prevalence estimates, cyberbullying could still be on the rise, especially following the COVID-19 pandemic, perhaps due to students’ increased technology use [10], which also has the potential to increase adolescents’ risk of experiencing psychological, behavioral, and health problems associated with the involvement in such phenomena [2,11,12,13].

A recent systematic review found a strong negative association between cyberbullying and mental health outcomes in youth, suggesting the pervasive impact of digital harassment on individual well-being [14]. Meta-analytic studies have also demonstrated that, regardless of the role of cyberbullies and/or cybervictims, being involved in cyberbullying has pervasive and negative outcomes. Both cyberbullying and cybervictimization were associated with internalizing and externalizing outcomes, including involvement in delinquency and violent and deviant behaviors [11,15,16,17,18,19,20,21,22,23].

Considering the social alarm due to the increasing spread of cyberbullying over time, numerous cyberbullying prevention and intervention programs have been implemented, evaluating their efficacy. While some pre-existing school bullying or school violence programs have been extended to include modules on cyberbullying, other programs have been developed to target cyberbullying and cybervictimization [24].

Comparisons across cyberbullying intervention and prevention programs’ features, activities, duration, and targets are often complex.

In this regard, in recent meta-analytic research, Lan et al. [25] categorized 19 anti-cyberbullying programs according to their components and features into five main pillars: (1) programs focusing on student peer tutoring and knowledge mobilization (SPTKM); (2) programs focusing on students’ knowledge mobilization (SKMTad); (3) programs focusing on teacher adaptation (Tad); (4) programs focusing on instruction-centered information (ICIS) and (5) programs focusing on student peer tutoring and community (SPT + com). The results concerning such programs’ efficacy in preventing and reducing cyberbullying underlined that agency-oriented programs targeting students and teachers (SPTKM, SKMTad, and Tad) are significantly more effective than information support-oriented (ICIS and SPT + com).

Polanin et al. [24], alongside other meta-analytic studies [26,27], reported that most prevention programs proved effective in reducing cyberbullying and cybervictimization. Conversely, a recent systematic review reported inconsistent results regarding programs’ effectiveness [28]. In addition, as Ng et al. [29] stressed, follow-up assessment varied widely across studies, ranging from 5 weeks to 1.5 years.

However, the extant research agrees on the absence of a long-term evaluation of anti-cyberbullying programs, underlining that evidence concerning their efficacy in reducing cyberbullying behaviors over time remains scarce [24,25,26,30,31].

To the best of our knowledge, only five studies have evaluated the long-term effectiveness of anti-bullying programs (see Table 1): the Cyber Friendly Schools program [32], NoTrap! [33], the Visc Social Competence Program [34], PREDEMA [35], and the Learning Together Program [36].

The Learning Together Program [36] is designed to improve the school environment by adopting a restorative approach. Implemented in England, the program lasted 36 months, and its efficacy assessment reported contrasting results. While the 24-month follow-up showed a significant reduction for only cybervictimization, in the long-term (36-month follow-up), the program effectively reduced cyberbullying only. The authors explained such findings as results of the chance, underlining that the program was not developed to target cyberbullying and cybervictimization specifically. Furthermore, intervention activities were delivered with variable intensity during all 36 months of the program implementation, thus making it impossible to assess the program’s sustainability and duration over time.

In Spain, the PREDEMA [35] socio-emotional intervention program aimed at preventing cyberbullying and improving subjective well-being was assessed concerning its efficacy. It was proven effective in reducing cyberbullying and cybervictimization immediately after the intervention and at a 6-month follow-up. The program’s long-term efficacy was explained by highlighting the program’s focus on participants’ training, empowerment, and consolidation of socio-emotional abilities, such as emotional regulation and management.

The Visc Social Competence Program [34] is a one-year-long primary and secondary prevention program for teachers and students implemented in Austria. The program aims to deliver a cascading school-wide training by teaching social–emotional skills emphasizing traditional bullying. Students in the experimental schools reported a diminished level of cyberbullying at the 6-month follow-up compared to the participants in the control schools, which showed increasing cyberbullying and cybervictimization behaviors over time. The program reduced cyberbullying and cybervictimization after one year, even if the long-term follow-up assessment involved a sub-sample of only six schools compared to the initial sample of 18. Similar to Shoeps et al. [35], it is possible to speculate that the consolidation of some socio-emotional skills takes a long time, becoming evident in the long run rather than in a few months, resulting in a significant reduction in both cyberbullying and cybervictimization after one year rather than after 6 months.

In Australia, the Cyber Friendly School (CFS) program [32] involved the entire school community in empowering teachers, parents, and students to prevent and reduce cyberbullying through online materials and face-to-face activities. After 18 months, the experimental group showed a significant decrease in cyberbullying and cybervictimization. However, the long-term follow-up indicated no significant differences in target behaviors between the experimental and control groups.

Lastly, the third edition of the Italian prevention program NoTrap! [33] provided data on its long-term efficacy in preventing and reducing cyberbullying and cybervictimization. The NoTrap! is a peer-led program including face-to-face and online training and activities. Results showed a significant decrease in cyberbullying and cybervictimization behaviors, which remained stable over time (1-year follow-up).

Although changes between pre-test and post-test assessments (i.e., program effectiveness) have been extensively studied, only a few empirical studies have evaluated the ongoing impact of such programs with long-term follow-up assessments [24,25,26,29,30,33,37]. Investigating cyberbullying prevention and intervention programs’ sustainability and over time efficacy would be crucial to understanding and evaluating which programs’ components and features work better in terms of effective primary and secondary prevention strategies to protect children and youth from the involvement in cyberbullying and cybervictimization over time [25,29,30].

### The Present Study

Although extant literature provides evidence of the effectiveness of cyberbullying prevention programs in the short term, their long-term sustainability remains unclear. There is a dearth of empirical research, including follow-up over the long run for these programs. Therefore, to address such a gap in the existing literature, the present study examines the long-term effects of the Tabby Improved Prevention and Intervention Program (TIPIP) [38] on cyberbullying and cybervictimization.

TIPIP is a theoretically based multi-component program. Combining Bronfenbrenner’s Ecological System Theory [39] and the Threat Assessment Approach [40], the program focuses on the assessment and management of significant risk factors for cyberbullying and cybervictimization [41], considering and targeting significant adults (teachers and parents) and peers in training, cooperative and in group activities, adhering to the critical criteria for building effective cyberbullying prevention and intervention programs [6]. While the Threat Assessment Approach [40] was used to understand how to best pre-vent a threat for antisocial behaviors to occur, the Ecological Systems Theory [39] was adopted for identifying the levels (individual, interpersonal, social, community) in which threating behaviors can be found, increasing the risk of aggressive behaviors involvement. The Tabby Improved program was developed for twofold aims: (1) identifying risk factors for cyberbullying and cybervictimization, and (2) identifying the ecological levels in which these factors operate and interact with each other.

Four main components constitute the TIPIP program: (1) training activities with teachers; (2) school conferences with parents; (3) online materials for students, teachers, and parents (i.e., Tabby toolkit; available at www.tabby.eu (accessed on 10 February 2023)); and (4) in-class activities with students.

Teacher training activities consist of three sessions (3 h each) delivered once a week plus an additional day on the possible civil, criminal, and administrative, legal implications of cyberbullying and the age of responsibility.School conferences with parents aim to: (i) inform parents about the prevention and intervention program activities and aims; (ii) sensitize and inform them about the cyberbullying problem and how to protect their children by setting clear rules about internet use and how to monitor their online activities best.The third component of the program is the Tabby “toolkit” [38]. The toolkit includes three elements. (i) First, there is a digitalized self-report questionnaire (the Tabby Improved checklist) used to measure risk factors for students’ involvement in cyberbullying and cybervictimization. (ii) Second, four short videos are used as stimuli to make youngsters think about the cyberbullying phenomenon and its consequences. The central theme in each video is the idea that there is always an alternative. Indeed, at the end of each video, the story ‘rewinds’, showing what would or could have happened if the character(s) in the video had opted for another alternative (desirable) possible choice. (iii) Third, there is a manual for teachers, parents and students containing useful evidence-based information on cyberbullying; it also includes a guide for trained teachers for them to organize class groups’ activities to raise students’ awareness about cyberbullying and cybervictimization.In-class activities with students consist of four sessions (2 h each) for each of the experimental classes. (i) First, group work is used to negotiate a shared definition of jokes, cyberbullying, and aggression. (ii) Next, students watch the Tabby toolkit videos. The videos were used as a stimulus to start a guided discussion regarding students’ experiences in cyberspace. (iii) Afterwards, group work was used to prepare at least ten rules or tips on avoiding risky online behaviors and involvement in cyberbullying and/or cybervictimization. (iv) Lastly, the fourth session focused on facing the legal consequences of cyberbullying. During this last session, a young boy who had been a cyberbully in the past met all experimental classes to share his story and explain his point of view, answer questions and discuss what made him realize the damage caused by his actions and what he is doing to address it to change. Further details on the program are available at [38].

TIPIP effectively reduced cyberbullying and cybervictimization immediately after the intervention implementation [38]. However, the study’s exploratory nature generated no hypotheses about the long-term program’s effectiveness.

## 2. Materials and Methods

### 2.1. Participants and Procedure

Participants were middle and high school students recruited from a convenient sample of 5 schools in the Campania region, South of Italy. Of the 49 classes involved, 20 (40.8%) were randomly allocated to the Experimental Group (EG, receiving the TIPIP intervention), while 29 (59.2%) were in the Control Group (CG, not receiving any intervention). None of the schools agreed to participate as a pure control school. Hence, classes were randomly assigned either to EC or to CG by the first author, in order to avoid possible teacher selection bias or class bias. Possible contaminations effects were controlled using intraclass correlation coefficients (see data analysis for more details); moreover, the first author coordinated and delivered all the activities that involved participants in the EG.

Data were collected from December 2015 to December 2016 at 3-time points: at baseline (T1), immediately after the delivery of the intervention (6 months from the baseline, T2), and after 1 year from the baseline measure (T3). At Tl, the sample consisted of 759 students (M = 47.9%) aged between 10 and 17 years (*M_age_* = 12.20, SD = 1.46). At T2, the sample consisted of 622 (*M_age_* = 12.56, SD = 1.48; M = 48.1%). At T3, the sample consisted of 475 students (*M_age_* = 13.36; SD = 1.42; M = 49.3%).

A total of 475 students participated in all three measurement points over the year. Students were invited to participate and enrolled in the study after obtaining informed parental consent. During regular school hours, students were asked to complete an online self-report questionnaire containing questions about their use of new communication technologies, referring to the previous six months. Before the administration, the first author instructed participants about the meaning of the term cyberbullying, to ensure they had a common understanding of the main topic of the questionnaire. The following definition was provided: “Cyberbullying is an aggressive and intentional act, carried out by a group or an individual, using electronic forms of contact, repeatedly over time against a victim who cannot easily defend himself/herself” (p. 376) [4].

A unique code was created by each student and matched throughout the study’s stages (Participants were instructed on how to generate their personal code, following these instructions: “Insert your personal code (two numbers of your date of birth- for example, 03 if you were born on the 3rd, the last two letters of your surname, and the last 3 numbers of your mobile or home phone number/if you don’t have it, e.g., 03BA362, for Barbara born on the 3rd, with mobile nr: + + 362).).

All study procedures were conducted by the guidelines of the Declaration of Helsinki and its later amendments [42]. In addition, before the data collection, the approval of the Department of Psychology’s Ethical committee (29/2015) was obtained.

### 2.2. Measures

The TABBY Improved checklist [38,43] is a self-report measure designed to assess risk factors for youngsters’ involvement in cyberbullying and cybervictimization, after reviewing the international literature available [41], also according to the ecological theoretical framework [39] and the Threat Assessment Approach [40].

The checklist consists of 12 scales for a total of 130 items. Although participants completed the entire measure, for the present study, we used only two scales: involvement in cyberbullying and cybervictimization (5 items each; e.g., “I disclosed online private information or images without the person’s consent”, “I was actively engaged in excluding someone from an online group”). According to the taxonomy proposed by Willard [44], participants indicate their involvement, in the previous six months, in the following behaviors: flaming (sending violent/vulgar online messages), denigration, impersonation, outing, and exclusion. Each item is rated on a five-point Likert scale from 0 (“It has never happened in this period”) to 4 (“It happened several times a week”).

In the present study, Cronbach’s alpha ranged between 0.64 and 0.84 for cyberbullying and 0.72 and 0.77 for cybervictimization throughout the three-time points.

### 2.3. Data Analysis

All statistical analyses were conducted using the IBM Statistical Package for the Social Sciences (version 26) [45].

Firstly, we conducted attrition analyses to examine potential bias between participants who had completed measures across all time points and participants who dropped out at times 2 and/or 3. Results indicated no statistically significant differences in the study’s variables (cyberbullying: *F*_(1,757)_ = 2.300, *p* = ns; cybervictimization: *F*_(1,757)_ = 0.11, *p* = ns) between retained participants and those who dropped out. Students’ dropout was mainly due to absence on data collection days and mistakes in personal ID creation. Therefore, the dataset was not likely to be biased due to attrition, suggesting that the main effects should be considered stables.

Preliminary descriptive analyses were conducted to elucidate the students’ demographic and Internet use characteristics in the sample.

Afterward, we compared participants between EG and CG at baseline about cyberbullying and cybervictimization using ANOVA statistics.

Given the nested structure of the sample (participants nested in classes and schools), we examined the intraclass correlation coefficients (ICCs) to measure the strength of the association among EG and CG groups repeatedly measured at baseline and follow-up stages. The ICCs were generally small—under the 0.05 value proposed for clustering design [45]—and the analysis was expected to be unaffected by clustering effects.

To evaluate the within-group changes in cyberbullying and cybervictimization over time (long-term program effectiveness) we used two 2 × 3 mixed ANCOVAs for longitudinal analysis, with group as the between-subjects factor (EG vs CG) and time as the within-subjects factor (pre- and 2 post-tests). Participants’ gender was introduced as covariate.

A *p*-value < 0.05 was considered statistically significant for all analyses.

## 3. Results

### 3.1. Descriptive Statistics

The sample consisted of 475 students, of whom 167 were in the EG (*M_age_* = 13.15; SD = 1.52; M = 51.5%) and 308 were in the CG (*M_age_* = 13.47; SD = 1.35; M = 47.7%).

Out of the sample, 65.3% of students indicated having more than one social network profile; among them only 9.4% reported knowing half of their online contacts. About one-third of the students (38.2%) spend between 2 and 4 h online, and 16.4% indicated that their parents had never given them clear rules about Internet use. Similarly, 24.4% have parents who do not monitor their online activities (for more details, see Table 2).

Regarding cyberbullying and cybervictimization experiences, 15.8% indicated they had been perpetrators, while 29.0% were cybervictimized.

However, no differences emerged between the EG and CG groups regarding cyberbullying (F_(6)_ = 0.56, *p* > 0.05) and cybervictimization (F_(15)_ = 1.67, *p* > 0.05) at the baseline.

Furthermore, Figure 1 and Figure 2 report average scores of cyberbullying and cybervictimization at baseline (T1, N = 759), after delivering the intervention (T2, N = 622), and at follow-up (T3, N = 475) assessed separately for EG and CG

### 3.2. Long-Term Effect of Tabby Improved Prevention and Intervention Program (TIPIP)

Examining the long-term effect of the program on cyberbullying, the mixed ANCOVA with a Greenhouse–Geisser correction showed a non-significant group effect (*F*_(1,442)_ = 0.063, *p* = 0.803, η^2^ = 0.000), a non-significant time*gender interaction (*F*_(2,441)_ = 0.381, *p* = 0.683, η^2^ = 0.002) and a non-significant group*time interaction (*F*_(2,441)_ = 0.337, *p* = 0.683, η^2^ = 0.002). A similar pattern of results was found for cybervictimization, mixed ANCOVA’s results indicated a non-significant group effect (*F*_(1,442)_ = 0.558, *p* = 0.455, η^2^ = 0.001), a non-significant group*time interaction (*F*_(2,441)_ = 0.821, *p* = 0.441, η^2^ = 0.004) and a non-significant time*gender interaction (*F*_(2,441)_ = 0.082, *p* = 0.921, η^2^ = 0.000). However, looking at average scores (Table 3) over time, there is a decrease both in CB and CV at T2 for students in the EG.

## 4. Discussion

The main aim of the present study was to evaluate the long-term effect of the Tabby Improved Prevention and Intervention Program [38] for cyberbullying and cybervictimization in a sample of middle and high school Italian students.

Research on cyberbullying [6,13,28,37,46,47] has supported the development and implementation of multi-component anti-cyberbullying educational programs with a solid theoretical framework. Since the pervasive nature of cyberbullying and its enormous impact on youth mental health [48] and the social and public health burden [8], primary prevention activities should adopt a wider approach in order to target, as well as cyberbullying and cybervictimization, other phenomena linked to peer aggressive behaviors. 

Prevention and early intervention are key elements for improving individual health and well-being and minimizing adverse consequences [49]. Thus, several anti-cyberbullying programs have been developed and assessed for their short-term efficacy [24,25,26,27,29,31]. To the best of our knowledge, only a few programs have been evaluated regarding sustainability and duration over time, reporting contrasting results. In this regard, a recent systematic review [6] provides critical criteria for building effective cyberbullying prevention and intervention programs: (1) a theory that serves as the foundation for all strategies; (2) a focus on risk and protective factors identified in the empirical literature; (3) the importance of various contexts affecting an individual (e.g., home, school, etc.) in program design; and (4) empirical evaluation of program outcomes.

Even though the TIPIP [38] contained all the above-reported criteria, the results concerning its long-term efficacy demonstrate a non-significant impact of the program at the 1-year follow-up.

Consistent with the literature reviewed [24,25,26,27,31], our results demonstrated that our program is ineffective in tackling cyberbullying and cybervictimization in the long term. Although the TIPIP was associated with significantly decreasing cyberbullying and cybervictimization at the post-implementation follow-up (T2) [38], our mixed ANCOVAs do not indicate significant changes across three time points. Speculatively, it could be hypothesized that such results are due to students’ dropouts between T2 and T3, which have altered the effects over time point. However, this result suggest that TIPIP’s long-term effectiveness was limited.

At 1-year follow-up, there was evidence that cybervictimization was lower among the CG than the EG.

Although no iatrogenic effect could be supposed since both groups reported an increased value at the 1-year follow-up, such a pattern of results proves that the program is ineffective in maintaining expected changes over the long run.

Considering the promising results concerning the long-term efficacy of some cyberbullying prevention and intervention programs, such as NoTrap [33] and PREDEMA [35], it could be useful to include specific modules on social–emotional learning (SEL) and peer training and mentoring activities within preventive programs for cyberbullying [50]. Since different patterns of individual, relational, and contextual risk factors are associated with the onset of involvement in cyberbullying and cybervictimization [51], different curricula should be implemented to prevent and manage perpetration and victimization.

In this regard, programs including specific modules aimed at students’ empowerment of socio-emotional competencies such as emotion regulation and management of interpersonal conflicts and including peer education mentoring activities have been found effective in reducing cyberbullying, while, activities that aimed to increase youth awareness of online risks were found to be more effective in contrasting and preventing cybervictimization [50].

Drawing from these assumptions, a comprehensive program addressing the psychological characteristics and mechanisms related to violent and aggressive behaviors and risk factors associated with cyberbullying and cybervictimization among youth has the potential to be an effective preventive initiative. Thus, future anti-cyberbullying programs, including the TIPIP, should be enlarged and integrated by including both activities focusing on psychological mechanisms and processes (i.e., self-esteem, emotion regulation abilities) and assessing and managing risk factors associated with the involvement in cyberbullying and cybervictimization.

### Limitations and Future Directions

There were also several limitations that must be acknowledged. Firstly, the study relies exclusively on self-report measures, which may be biased by common method variance [52]. Future studies should include a multi-method (e.g., interview) and multi-informant (e.g., teachers, peers) assessment in order to obtain a more reliable and comprehensive evaluation. Secondly, the EG was a relatively small sample of Italian youth, so the results might not generalize to other samples with different school systems and cultural contexts. Future research should also attempt to replicate our findings, including a larger and more general student body population. Thirdly, the study may be vulnerable to cross-contamination between experimental and control participants. However, every effort was made to control for contamination. Fourthly, the program was implemented about seven years ago, so data could appear outdated. However, considering the few existing studies concerning anti-cyberbullying programs’ long-term efficacy and sustainability, we believe that our results shed light on the importance of evaluating them. Moreover, the results of the current investigation stress the importance for future studies to investigate which components or activities effectively work over time in reducing cyberbullying and cybervictimization. Fifthly, even though the first author continually checked with teachers and parents regarding their comprehension of the program’s contents throughout the intervention phase, adults’ adherence to the intervention was not systematically assessed. Future studies should complement such programs with some measure of treatment integrity and address all its five dimensions (i.e., adherence, exposure, program differentiation, quality of delivery, and participant responsiveness) [53,54,55].

Lastly, the reliability value for cyberbullying was relatively low (around 0.60) and required caution. However, the value is deemed acceptable given the short-scale dimension [56]. Despite the above limitations, our findings expand the limited research about the long-term effect of anti-cyberbullying programs and propose a new direction for prevention.

## 5. Conclusions

Considering that according to a meta-analytic review, few anti-cyberbullying programs have been assessed for their efficacy over time, the present study aimed to evaluate the long-term effectiveness of the TIPIP, assessing its sustainability in reducing cyberbullying and cybervictimization after one year from the baseline. Evaluating such programs’ long-term effects and sustainability in preventing and reducing cyberbullying and cybervictimization could be a turning point for developing effective primary and secondary prevention programs.

In fact, despite our results underlying the TIPIP inefficacy in the long-term assessment, such studies contributed to increasing our knowledge about what works in preventing and reducing cyberbullying and cybervictimization.

From this perspective, future anti-cyberbullying programs should be developed or integrated to include modules and activities proven effective in the long run in contrasting adolescents’ involvement in cyberbullying and cybervictimization, thus making possible cross-cultural comparisons possible and the development and the implementation of transnational effective strategies and policies.

## Figures and Tables

**Figure 1 ijerph-20-05436-f001:**
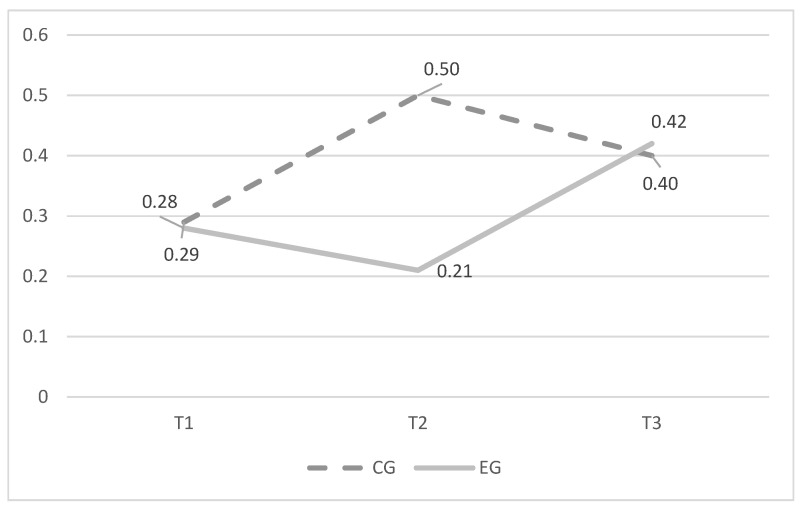
Average scores in cyberbullying over time.

**Figure 2 ijerph-20-05436-f002:**
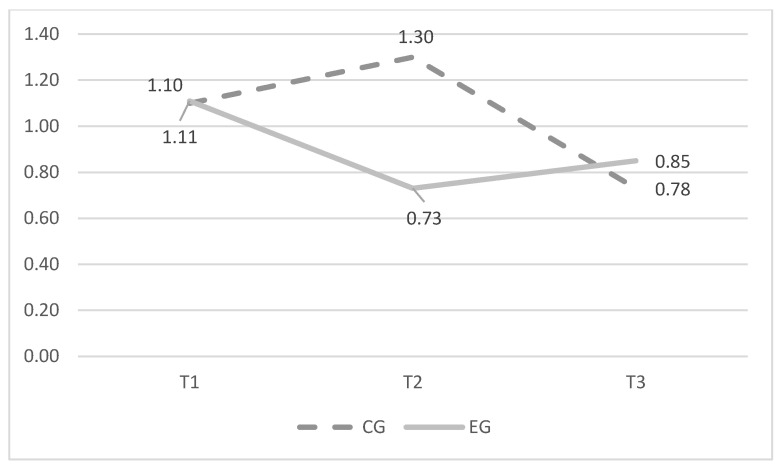
Average scores in cybervictimization over time.

**Table 1 ijerph-20-05436-t001:** Overview of anti-cyberbullying programs with a long-term follow-up.

Study	Related Program Element	Sample		Measurement	Results	
		N	Age		Post Intervention (T1)	Long-Term Follow-Up (T2)
Learning Together [36] England	Social–emotional learning = Yes Whole-school approach = Yes Peer mentoring = No Education on cyberbullying and online safety = No	6.667 (47.3% M) EG = 3.320 CG = 3.347	11–12	Duration: 36 months T1 = 24 months after the baseline (T0) T2 = follow-up 12 months after T1.	EG only showed lower CV rates than the CG.	EG reduced CB but not CV.
PREDEMA [35] Spain	Social–emotional learning = Yes Whole-school approach = No Peer mentoring = No Education on cyberbullying and online safety = No	N = 360 - EG = 168 CG = 192	-	Duration: 9 months T1 = intervention assessment after 3 months form the baseline (T0) T2 = follow-up test 6 months after T1).	EG significantly scored lower in CB and CV compared to CG.	The reduction in CB and CV among students of the EG remained stable even 6 months later.
ViSC Social Competence Program [34] Austria	Social–emotional learning = Yes Whole-school approach = Yes Peer mentoring = No Education on cyberbullying and online safety = No	N = 1.639 (52.4% M) EG = 1.192 CG = 447	10–15	Duration = 18 months T1 = after intervention assessment one year later the baseline (T0) T3 = follow-up test 6 months after T2.	EG remained relatively stable CG had a rise in CB and CV.	EG decreased their involvement in CB and CV. CG experienced an increase in both CB and CV
Cyber Friendly Schools [32] Australia	Social–emotional learning = No Whole-school approach = Yes Peer mentoring = Yes Education on cyberbullying and online safety = Yes	N = 3.382 (47.0% M) EG = 1.878 CG = 1.054	M_age_ = 13	Duration: 3 years T1 = 18 months after the baseline (T0) T2 = follow-up test 1 year after T1.	Reduction both CB and CV in EG.	No significant differences in CB e CV between the EG and CG
No Trap! [33] Trial 1 Italy	Social–emotional learning = Yes Whole-school approach = Yes Peer mentoring = Yes Education on cyberbullying and online safety = Yes	N = 622 (60.3% M) EG = 451 CG = 171	14–18	Duration = 1 year T1 = after intervention assessment 6 months after the baseline (T0) T2 = follow-up test 6 months after T1.	EG showed a significant decrease in both cybervictimization, and cyberbullying compared to CG	EG reduction in both cybervictimization, and cyberbullying was stable at T2.

Note: EG = Experimental Group, CG = Control Group, CB = Cyberbullying, CV = Cybervictimization.

**Table 2 ijerph-20-05436-t002:** Descriptive statistics of the sample.

		Overall (N = 475)	EG (*n* = 167)	CG (*n* = 308)
Age		M = 13.36 (SD = 1.42)	13.15 (SD = 1.52)	13.47 (SD = 1.35)
Gender		49.3% M	51.5% M	47.7% M
Presence of social network profile(s)	More than one	65.3%	63.5%	66.2%
Personally know friends on social network	Only half	9.4%	11.3%	8.4%
Parents talk with students about Internet safety	Never	9.3%	10.2%	8.8%
Parents monitor students’ online activities	Never	24.4%	26.3%	23.4%
Parents giving rule concerning internet use	Never	16.4%	14.4%	17.5%
Hours per day online	0–1	24.8%	21.6%	26.6%
	2–4	38.3%	38.3%	38.3%
	6–8	21.1%	16.2%	23.7%
	10–12	9.1%	14.4%	6.2%
	>12	6.7%	9.6%	5.2%
Cyberbullying	At least once	14.8%	15.6%	14.4%
Cybervictimization	At least once	29.0%	34.1%	26.1%

**Table 3 ijerph-20-05436-t003:** Mean and standard deviations of cyberbullying and cybervictimization across conditions over time.

		T1	T2	T3
		M (SD)	M (SD)	M (SD)
	Cyberbullying			
EG		0.35 (0.89)	0.29 (0.71)	0.42 (1.81)
CG		0.33 (1.24)	0.37 (1.52)	0.40 (1.56)
	Cybervictimization			
EG		1.22 (2.15)	0.95 (1.64)	0.85 (2.17)
CG		0.97 (1.95)	0.97 (2.17)	0.73 (1.77)

## Data Availability

The data presented in this study are available on request from the corresponding author.

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
