# Peer review of "The Long-Term Efficacy and Sustainability of the Tabby Improved Prevention and Intervention Program in Reducing Cyberbullying and Cybervictimization"

_ijerph, 2023, doi:10.3390/ijerph20085436_

Round 1
Reviewer 1 Report
I would like to thank you for this opportunity to review this interesting manuscript. The manuscript is well written and it addresses important and current topics.
Before I suggest accepting the paper I hope the authors could provide more details on the following issues listed below: The control group should be representative of the given student population. I hope they could provide more detail on the inclusion/exclusion (eligibility) criteria of the samples. How were the schools selected? It is not clear why authors decided to focus on merely cyberbullying/cyber victimization. For instance, is it not clear where the confounding factors were identified? I think that the method section should include a description of the intervention program in detail. This section could include description on components and structure of the program what aims each component is addressing. Regarding statistical analysis, I was wondering whether a linear mixed model could be useful for this purpose? Personally, I am not convinced of the use of ANOVA in this kind of concept. I hope the authors could consider this issue further and possibly provide more foundation to support their decision. Otherwise, I consider the manuscript to be promising and interesting. Especially, I like the description of the theoretical rationale.Author Response
Q1: I would like to thank you for this opportunity to review this interesting manuscript. The manuscript is well written and it addresses important and current topics.
R: We want to thank Reviewer 1 for this recognition and their valuable comments. We revised the manuscript accordingly, and revisions are highlighted using the MS Word track changes feature.
Q2: Before I suggest accepting the paper. I hope the authors could provide more details on the following issues listed below: The control group should be representative of the given student population. I hope they could provide more detail on the inclusion/exclusion (eligibility) criteria of the samples. How were the schools selected?
R: We thank Reviewer 1 for pointing out this issue. Regarding eligibility, no inclusion/exclusion criteria were considered for the sample selection. Schools’ participation was voluntary, and all students were invited to participate (please see lines 209-214).
Q3: It is not clear why authors decided to focus on merely cyberbullying/cyber victimization. For instance, is it not clear where the confounding factors were identified?
R: The present study is part of a larger project examining cyberbullying and cybervictimization among (pre)adolescents. Considering the aim of our work, which focused on the long-term efficacy and sustainability of an already implemented program (TIPIP program), we considered cyberbullying and cybervictimization scores over time across experimental and control groups. However, considering gender differences in such behaviors across participants’ gender, we changed our analytic plan. Instead of logistic regression analyses, we used two 2x3 mixed ANCOVAs for longitudinal studies, with group as the between-subjects factor (EG vs. CG) and time as the within-subjects factor (pre- and two post-tests). Participants’ gender was introduced as a covariate.
Q4: I think that the method section should include a description of the intervention program in detail. This section could include description on components and structure of the program what aims each component is addressing.
R: Thank you for your comment. We agree with the critical point you have raised here. Thus, we have reviewed the Method section, adding more details about the Tabby Improved program (lines 159-198).
Q4: Regarding statistical analysis, I was wondering whether a linear mixed model could be useful for this purpose? Personally, I am not convinced of the use of ANOVA in this kind of concept. I hope the authors could consider this issue further and possibly provide more foundation to support their decision.
R: We thank the reviewer for this suggestion. Considering the longitudinal nature of our study and in line with other studies (e.g., Shoeps et al., 2018; Palladino et al., 2016), we changed our analytic plan to perform two 2x3 mixed ANCOVAs for longitudinal analysis, with group as the be-tween-subjects factor (EG vs. CG) and time as the within-subjects factor (pre- and two post-tests). Participants’ gender was introduced as a covariate.
Q5: Otherwise, I consider the manuscript to be promising and interesting. Especially, I like the description of the theoretical rationale.
R: Many thanks again
Reviewer 2 Report
Thank you for the opportunity to review this paper. This study is an interesting manuscript; however, I have some concerns outlined below:
Highlight the knowledge gap, the significance of the study, and the contribution of the expected study findings to the knowledge base.
Have you received permission from the original authors to reuse their questionnaires?
The intervention sounds very interesting. However, the authors should be more explicit about the rationale of the intervention and how it was developed.
Given the conceptual and empirical interconnectedness of the DVs, I believe that MANOVA (together with individual ANOVAS for each DV) would be more appropriate. Furthermore, Bonferroni(-Holm) adjustment and information on effect sizes are necessary.
The following items need to be added to the Method section.
1) Details of whether and how the intervention was standardized.
2) Details of whether and how care providers’ adherence to the protocol will be assessed or enhanced.
3) Details of whether and how participants’ adherence to interventions will be assessed or enhanced.
Author Response
Q1: Thank you for the opportunity to review this paper. This study is an interesting manuscript; however, I have some concerns outlined below: Highlight the knowledge gap, the significance of the study, and the contribution of the expected study findings to the knowledge base.
R: Thank you very much for your feedback. We appreciate your consideration and time in reviewing the manuscript. It has been substantially improved thanks to the concerns you have raised.
Q2: Have you received permission from the original authors to reuse their questionnaires?
R: We thank Reviewer 2 for the comment. The Tabby Improved checklist is an online actuarial checklist developed thanks to the implementation of the European project (Baldry, Farrington & Blaya, 2018), which was improved by carrying out a narrative review on risk factors for cyberbullying and cybervictimization (Baldry et al., 2015), and then used for the assessment of the short- time efficacy of the Tabby Improved program (Sorrentino et al., 2018). The first author worked with Prof. Anna C. Baldry (who unfortunately passed away on 09 March 2019) on developing the above-mentioned checklist. Questions concerning cyberbullying and cybervictimization involvement are based on types of cyberbullying behaviors proposed by Willard (2007) (as reported in line 245). We now made it more clear that the questionnaire is based on Willard's taxonomy. The checklist is a new instrument created by the first author (i.e., lines 236-239).
Q3: The intervention sounds very interesting. However, the authors should be more explicit about the rationale of the intervention and how it was developed.
R: Thank you for your comment. We understand your viewpoint and apologize for any confusion. The Tabby Improved program is rooted in Bronfenbrenner’s Ecological System Theory (EST) and the Threat Assessment Approach. The EST (Bronfenbrenner, 1979) provides a comprehensive theoretical framework of the extent to which an individual’s involvement in cyberbullying and/or cybervictimization is affected by several factors: the students’ involvement, their families, peers, school, and community. The threat assessment approach (Fein et al., 1995) helps to recognize and evaluate the presence of those risk factors that the international literature suggests are significant for students’ involvement in cyberbullying and cybervictimization (Baldry et al., 2015). Thus, the Tabby Improved program has a sound theoretical background and was developed for identifying risk factors for cyberbullying and cybervictimization and the ecological levels in which those risk factors operate and interact with each other in the directing of assessing the future risk of any threatening circumstances (risk factors) taking place. As suggested, we have further elaborated the rationale (lines 154-166).
Q4: Given the conceptual and empirical interconnectedness of the DVs, I believe that MANOVA (together with individual ANOVAS for each DV) would be more appropriate. Furthermore, Bonferroni(-Holm) adjustment and information on effect sizes are necessary.
R: We thank the Reviewer for this suggestion. The present study is part of a larger project examining cyberbullying and cybervictimization among (pre)adolescents. Considering the aim of our work, which focused on the long-term efficacy and sustainability of an already implemented program (Tabby Improved Program), we considered cyberbullying and cybervictimazion scores over time across experimental and control groups. However, considering gender differences in such behaviors across participants ‘gender, we changed our analytic plan. Instead of logistic regression analyses, we used two 2x3 mixed ANCOVAs for longitudinal studies, with group as the between-subjects factor (EG vs. CG) and time as the within-subjects factor (pre- and 2 post-tests). Participants’ gender was introduced as a covariate.
Q5: The following items need to be added to the Method section.
1) Details of whether and how the intervention was standardized.
2) Details of whether and how care providers’ adherence to the protocol will be assessed or enhanced.
3) Details of whether and how participants’ adherence to interventions will be assessed or enhanced.
R: These issues have now been addressed in the detailed description of the procedure (i.e., lines 159-198; 209-214). However, adherence to the protocol was not checked using quantitative instruments. Thanks for pointing this out. We addressed this issue in the discussions section (i.e., lines 436-441).
Reviewer 3 Report
The article addresses a cognitively interesting and socially important issue. The authors have carried out sound research. Both the theoretical part and the description of the methodology and the research results have been conducted fairly. The analyses are thorough, supported by solid research. However, the research is outdated, as it is from 2015-2016. In terms of cyberbullying and cybervictimisation, a lot has changed in recent years. It would be useful to re-examine this issue and compare research findings.
The critical, yet substantiated, comments on the Tabby Improved Prevention and Intervention (TIPIP) in Reducing Cyberbullying and Cybervictimisation programme are valuable. The findings deserve the attention not only of researchers, but also of practitioners - teachers, educators, parents.
Author Response
Q1: The article addresses a cognitively interesting and socially important issue. The authors have carried out sound research. Both the theoretical part and the description of the methodology and the research results have been conducted fairly. The analyses are thorough, supported by solid research. However, the research is outdated, as it is from 2015-2016. In terms of cyberbullying and cybervictimisation, a lot has changed in recent years. It would be useful to re-examine this issue and compare research findings.
R: We thank Reviewer 3 for the feedback. We appreciate your consideration and time in reviewing the manuscript. We discussed the outdated nature of our data in the limitations section (i.e., lines 430-436).
Q2: The critical, yet substantiated, comments on the Tabby Improved Prevention and Intervention (TIPIP) in Reducing Cyberbullying and Cybervictimisation programme are valuable. The findings deserve the attention not only of researchers, but also of practitioners - teachers, educators, parents.
R: Thank you for your thoughtful and thorough review. We agree with the important point you have raised here. Thus, we have listed the outdated nature of our research in the limitation section (lines 430-436).
Round 2
Reviewer 2 Report
Thank you for your effort on the revised manuscript. Good Job!